# ENERGY-BASED OUT-OF-DISTRIBUTION DETECTION FOR MULTI-LABEL CLASSIFICATION

## ABSTRACT

Out-of-distribution (OOD) detection is essential to prevent anomalous inputs from causing a model to fail during deployment. Improved methods for OOD detection in multi-class classification have emerged, while OOD detection methods for multi-label classification remain underexplored and use rudimentary techniques. We propose *SumEnergy*, a simple and effective method, which estimates the OOD indicator scores by aggregating energy scores from multiple labels. We show that *SumEnergy* can be mathematically interpreted from a joint likelihood perspective. Our results show consistent improvement over previous methods that are based on the maximum-valued scores, which fail to capture joint information from multiple labels. We demonstrate the effectiveness of our method on three common multi-label classification benchmarks, including MS-COCO, PASCAL-VOC, and NUS-WIDE. We show that *SumEnergy* can reduce the FPR95 by up to 10.05% compared to the previous best baseline, establishing state-of-the-art performance.

## 1 INTRODUCTION

Out-of-distribution (OOD) detection is central for reliably deploying machine learning models in open-world environments, where new forms of test-time data may appear that were nonexistent during the training time. The problem of OOD detection has gained significant research attention lately, given its importance for safety-critical applications such as unseen disease identification (Cao et al., 2020). However, recent studies have primarily focused on detecting OOD examples in multi-class classification, where each sample is assigned to one and *only one* label (Bevandić et al., 2018; Hein et al., 2019; Hendrycks & Gimpel, 2016; Lakshminarayanan et al., 2017; Lee et al., 2018; Liang et al., 2018; Mohseni et al., 2020; Chen et al., 2020; Hsu et al., 2020; Liu et al., 2020). This can be restrictive in many real-world settings where images often have multiple objects of interest. For example, self-driving cars must differentiate between the road, traffic signs, and obstacles within a frame. In the medical domain, multiple abnormalities may be present in a medical image (Wang et al., 2017). Multi-label classification is desirable since there is no constraint on the number of classes an instance can be assigned to.

Currently, OOD detection in multi-label classification remains relatively underexplored. Out of the multi-label methods evaluated in (Hendrycks et al., 2019), MaxLogit achieved the best performance. However, simply using the maximum-valued logit is limiting because it does not incorporate information available from other possible labels. As seen in Figure 1, MaxLogit can only capture the difference between the dominant outputs for `dog` (in-dist.) and `car` (OOD), while positive information from another dominant label `cat` (in-dist.) is dismissed. Other baseline methods such as ODIN (Liang et al., 2018) and Mahalanobis (Lee et al., 2018) also derive scores based on the maximum score (*e.g.*, calibrated softmax score or Mahalanobis distance), and fail to capture the joint information. While energy scores have recently demonstrated superior OOD detection performance in the multi-class setting (Liu et al., 2020), this method does not trivially generalize to a multi-label classification setting where labels are not mutually exclusive. Hitherto, a key challenge lies in *how to leverage information across different labels*.

In this paper, we propose an energy-based method for OOD detection in the multi-label setting, which estimates OOD indicator scores jointly from multiple labels. We propose a simple and effective aggregation mechanism, *SumEnergy*, that seeks to combine the label-wise energy scores derived from individually independent classes. We show that the *SumEnergy* scores are mathematically

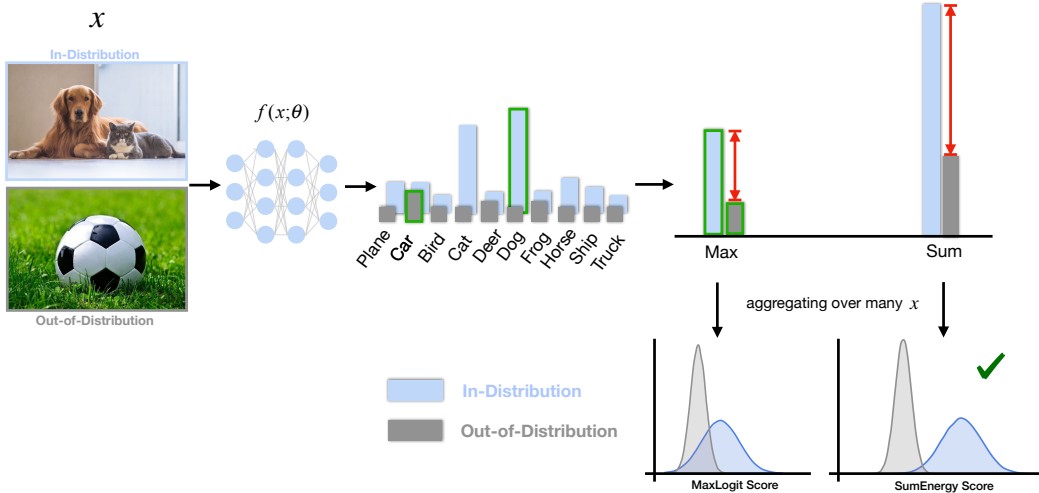

Figure 1: Energy-based out-of-distribution detection for multi-label classification. During inference time, input **x** is passed through classifier $f$, and label-wise scores are computed for each label. OOD indicator scores are either the maximum-valued score (denoted by green outlines) or the sum of all scores. Taking the sum results in a larger difference in scores and more separation between in-distribution and OOD inputs (denoted by red lines), resulting in better OOD detection. Plots in the bottom right depict the probability densities of maximum-valued versus summed scores.

meaningful, and can be interpreted from a joint likelihood perspective. Intuitively, an input with multiple dominant labels is more likely to be in-distribution, which is the key aspect that *SumEnergy* capitalizes on. As shown in Figure 1, summing label-wise energies over all labels amplifies the difference in scores between in-distribution and OOD inputs. Our method is parameter-free and can be conveniently used for any pre-trained multi-label classification model. Below we describe our contributions in detail.

**Contributions** First, we propose a theoretically motivated scoring function, *SumEnergy*, that is based on the aggregation over label-wise energy scores. Extensive experiments show that *SumEnergy* outperforms existing methods on three common multi-label classification benchmarks, establishing state-of-the-art performance. For example, on a DenseNet trained with MS-COCO (Lin et al., 2014), our method reduces the false positive rate (at 95% TPR) by **10.05**% when evaluated against ImageNet OOD data, compared to the best performing baselines. Consistent performance improvement is also observed on a different OOD test dataset Texture (Cimpoi et al., 2014a), as well as alternative network architecture.

Additionally, we perform a comparative analysis of how an alternative aggregation method affects OOD detection performance. In particular, we consider *MaxEnergy*, which takes the maximum energy score among the individual labels as the OOD indicator score. On a DenseNet trained with the PASCAL-VOC dataset, *MaxEnergy* yields 4.05% lower FPR95 compared to *SumEnergy*, which underlines the importance of taking into account scores derived from multiple labels.

Lastly, as an ablation, we demonstrate that energy scores are more compatible with the proposed summation aggregation method, compared to previous scoring functions such as logit (Hendrycks et al., 2019), MSP (Hendrycks & Gimpel, 2016), ODIN (Liang et al., 2018), and Mahalanobis distance (Lee et al., 2018). To see this, we explore the effectiveness of applying aggregation methods to previous popular scoring functions, which helps understand their applicability in the multi-label setting. We find that summing labels' scores using previous methods is inferior to summing labels' energies, emphasizing the need for *SumEnergy*. For example, simply summing over the logits across labels results in up to 51.93% degradation in FPR95 on MS-COCO, since the outputs are mixed with positive and negative numbers. In contrast, *SumEnergy* does not suffer from this issue because the signs of label-wise energy scores are uniform. More importantly, label-wise energy is provably aligned with the probability density of the corresponding label's training data. Our study therefore underlines the importance of properly choosing both the label-wise scoring function and the aggregation method. We show strong compatibility between the label-wise energy function and aggregation function, supported by both mathematical interpretation and empirical results.

## 2 METHODS

### 2.1 BACKGROUND: ENERGY FUNCTION IN MULTI-CLASS CLASSIFICATION

We consider a multi-class neural classifier $f(\mathbf{x}) : \mathbb{R}^D \to \mathbb{R}^K$, which maps an input $\mathbf{x} \in \mathbb{R}^D$ to K real-valued numbers as logits. A softmax function is used to derive a categorical distribution,

$$p(y_i \mid \mathbf{x}) = \frac{e^{f_{y_i}(\mathbf{x})}}{\sum_{j=1}^{K} e^{f_{y_j}(\mathbf{x})}}, \tag{1}$$

which indicates the probability for an input $\mathbf{x}$ to be of class $y_i$, with $i \in \{1, 2, ..., K\}$. A multi-class classifier can be interpreted from an energy-based perspective (Grathwohl et al., 2019) by viewing the logit $f_{y_i}(\mathbf{x})$ of class $y_i$ as an energy function $E(\mathbf{x}, y_i) = -f_{y_i}(\mathbf{x})$. Therefore, Equation 1 can be rewritten as:

$$p(y_i \mid \mathbf{x}) = \frac{e^{-E(\mathbf{x}, y_i)}}{\sum_{j=1}^{K} e^{-E(\mathbf{x}, y_j)}} \tag{2}$$

$$= \frac{e^{-E(\mathbf{x}, y_i)}}{e^{-E(\mathbf{x})}}. \tag{3}$$

By equalizing the two denominators above, we can express the *free energy* function $E(\mathbf{x})$ for any given input $\mathbf{x} \in \mathbb{R}^D$ in forms of:

$$E(\mathbf{x}) = -\log \sum_{i=1}^{K} e^{f_{y_i}(\mathbf{x})}. \tag{4}$$

### 2.2 ENERGY FUNCTION FOR MULTI-LABEL CLASSIFICATION

In this work, we propose using an energy-based method for OOD detection in multi-label classification networks, where an input can have several labels (see Figure 1). In what follows, we first introduce a label-wise energy function, and then propose aggregation methods that can leverage the joint information across labels in a theoretically meaningful way.

**Label-wise Free Energy** In multi-label classification, the prediction for each binary label $y_i$ is independently made by a binary logistic classifier:

$$p(y_i = 1 \mid \mathbf{x}) = \frac{e^{f_{y_i}(\mathbf{x})}}{1 + e^{f_{y_i}(\mathbf{x})}},$$

where $i \in \{1, 2, ..., K\}$. For brevity, we use $y_i$ in the probabilistic derivations to indicate the label being positive, *i.e.*, $y_i = 1$. The logistic classifier output can be viewed as the softmax with two logits—0 and $f_{y_i}(\mathbf{x})$, respectively. For each class $y_i$, we can define *label-wise free energy* as follows:

$$E_{y_i}(\mathbf{x}) = -\log(1 + e^{f_{y_i}(\mathbf{x})}). \tag{5}$$

**Aggregation Methods** We propose and contrast two aggregation mechanisms that seek to combine the label-wise energy scores derived above:

$$\textbf{Max} : \quad E_{\max}(\mathbf{x}) = \max_{i} -E_{y_i}(\mathbf{x}) \tag{6}$$

$$\textbf{Sum} : \quad E_{\text{sum}}(\mathbf{x}) = \sum_{i=1}^{K} -E_{y_i}(\mathbf{x}) \tag{7}$$

In particular, *MaxEnergy* finds the largest label-wise energy score among all labels; whereas *SumEnergy* takes the summation of energy scores across all labels. Note that in the above equations, label-wise energy $E_{y_i}(\mathbf{x})$ by definition is a negative value, and the aggregation methods output a positive value by negation.

**Mathematical Interpretation** We provide mathematical interpretations of different aggregation methods. First, by resorting to the energy-based model (LeCun et al., 2006), we show that the label-wise energy score is provably aligned with the conditional likelihood function. The conditional likelihood $p(\mathbf{x} \mid y_i)$ is given by:

$$p(\mathbf{x} \mid y_i) = \frac{e^{-E_{y_i}(\mathbf{x})}}{\int_{\mathbf{x}|y_i} e^{-E_{y_i}(\mathbf{x})}}, \tag{8}$$

where $Z_{y_i} = \int_{\mathbf{x}|y_i} e^{-E_{y_i}(\mathbf{x})}$ is the normalized density. Since $Z_{y_i}$ is the same with respect to all $\mathbf{x}$ with label $y_i$, the denominator in Equation 8 does not affect the overall distributional of label-wise energy scores. By taking the logarithm on both sides of Equation 8, we have:

$$E_{y_i}(\mathbf{x}) \propto -\log p(\mathbf{x} \mid y_i).$$

Given this, *MaxEnergy* can be interpreted as taking the maximum of conditioned log-likelihood among all labels:

$$E_{\max}(\mathbf{x}) \propto \max_i \log p(\mathbf{x} \mid y_i), \tag{9}$$

which does not take into account the joint information across labels.

In contrast, we provide mathematical interpretation for *SumEnergy*, which is the first method to consider the joint estimation of OOD scores across labels. We show that *SumEnergy* can be interpreted from the joint likelihood perspective:

$$E_{\text{sum}}(\mathbf{x}) = \sum_{i=1}^{K} \log \left( p(\mathbf{x} \mid y_i) \cdot Z_{y_i} \right) \tag{10}$$

$$= \sum_{i=1}^{K} \log p(\mathbf{x} \mid y_i) + \underbrace{\sum_{i=1}^{K} \log Z_{y_i}}_{Z:\text{constant for all } \mathbf{x}} \tag{11}$$

By applying Bayesian rule for each term $\log p(\mathbf{x} \mid y_i)$ in Equation 11, we have

$$E_{\text{sum}}(\mathbf{x}) = \log \prod_{i=1}^{K} \frac{p(y_i \mid \mathbf{x}) \cdot p(\mathbf{x})}{p(y_i)} + Z \tag{12}$$

$$= \log \prod_{i=1}^{K} p(y_i \mid \mathbf{x}) + K \cdot \log p(\mathbf{x}) + \underbrace{(Z - \log \prod_{i=1}^{K} p(y_i))}_{C: \text{ constant for all } \mathbf{x}} \tag{13}$$

Given all label $y_i$ are conditionally independent, we have $\prod_{i=1}^{K} p(y_i \mid \mathbf{x}) = p(y_1, y_2, ..., y_K \mid \mathbf{x})$. Therefore, Equation 13 is equivalent to:

$$E_{\text{sum}}(\mathbf{x}) = \log p(y_1, y_2, \ldots, y_K \mid \mathbf{x}) + K \cdot \log p(\mathbf{x}) + C \tag{14}$$

$$= \log \frac{p(\mathbf{x} \mid y_1, y_2, ..., y_K) \cdot \prod_{i=1}^{K} p(y_i)}{p(\mathbf{x})} + K \cdot \log p(\mathbf{x}) + C \tag{15}$$

$$= \underbrace{\log p(\mathbf{x} \mid y_1, y_2, ..., y_K)}_{\text{joint conditional log likelihood}} + \underbrace{(K-1) \cdot \log p(\mathbf{x})}_{\text{log data density}, \uparrow \text{ for in-distribution}} + \underbrace{Z}_{\text{constant for all } \mathbf{x}} \tag{16}$$

The equation above suggests that $E_{\text{sum}}(\mathbf{x})$ is in fact linearly aligned with the joint conditional log likelihood and log data density. The second term is desirable for OOD detection since it is aligned with the underlying data density, which is higher for in-distribution data $\mathbf{x}$ and vice versa. The first term takes into account joint estimation across labels, which is new to our multi-label setting and was not previously considered in multi-class setting (Liu et al., 2020). The first term allows even further discriminativity between in- vs. OOD data, since OOD data is expected to have lower joint conditional likelihood (*i.e.*, not associated with any of the labels). In contrast, having multiple

dominant labels is indicative of an in-distribution input, which is a characteristic that *SumEnergy* captures. More importantly, our method does not require estimating the density $Z$ explicitly, as $Z$ is sample independent and does not affect the overall *SumEnergy* score distribution.

We note that our derivation is based on the assumption that all labels are independent, which is in accordance with standard multi-label classification loss by treating each label as a binary prediction problem (Tsoumakas & Katakis, 2007). We consider this setting for its simplicity and generality. Our work also opens up an interesting future direction of OOD detection by considering the structural dependency among labels (Chen et al., 2017; 2019b;c).

## 2.3    AGGREGATED ENERGY FOR MULTI-LABEL OOD DETECTION

We propose using the aggregated energy functions $E(\mathbf{x})$ defined in Section 2.2 for OOD detection:

$$G(\mathbf{x}; \tau) = \begin{cases} \text{out} & \text{if } E(\mathbf{x}) \leq \tau, \\ \text{in} & \text{if } E(\mathbf{x}) > \tau, \end{cases} \tag{17}$$

where $\tau$ is the energy threshold, and can be chosen so that a high fraction of in-distribution data is correctly classified by $G(\mathbf{x}; \tau)$. $E(\mathbf{x})$ could take on forms of *Max* or *Sum*. A data point with higher aggregated energy $E(\mathbf{x})$ is considered as in-distribution, and vice versa (see Figure 1). We explore and provide the tradeoff of different aggregation methods in Section 3.2.

## 3    EXPERIMENTS

In this section, we describe our experimental setup (Section 3.1) and demonstrate the effectiveness of our method on several OOD evaluation tasks (Section 3.2). We also conduct extensive ablation studies and comparative analysis that leads to an improved understanding of different methods.

### 3.1    SETUP

**In-distribution Datasets** We consider three multi-label datasets: MS-COCO (Lin et al., 2014), PASCAL-VOC (Everingham et al., 2015), and NUS-WIDE (Chua et al., 2009). MS-COCO consists of 82,783 training, 40,504 validation, and 40,775 testing images with 80 common object categories. PASCAL-VOC contains 22,531 images across 20 classes. NUS-WIDE includes 269,648 images across 81 concept labels. Since NUS-WIDE has invalid and untagged images, we follow the work (Zhu et al., 2017) and use 119,986 training images and 80,283 test images.

**Training Details** We train three multi-label classifiers, one for each dataset above. The classifiers have a DenseNet-121 backbone architecture, with a final layer that is replaced by 2 fully connected layers. Each classifier is pre-trained on ImageNet-1K and then fine-tuned with the logistic sigmoid function to its corresponding multi-label dataset. We use the Adam optimizer (Kingma & Ba, 2014) with standard parameters ($\beta_1 = 0.9$, $\beta_2 = 0.999$). The initial learning rate is $10^{-4}$ for the fully connected layers and $10^{-5}$ for convolutional layers. We also augmented the data with random crops and random flips to obtain color images of size $256 \times 256$. After training, the mAP is 87.51% for PASCAL-VOC, 73.83% for MS-COCO, and 60.22% for NUS-WIDE.

**Out-of-distribution Datasets** To evaluate the models trained on the in-distribution datasets above, we follow the same set up as in (Hendrycks et al., 2019) and use ImageNet (Deng et al., 2009) for its generality. Besides, we evaluate against the Textures dataset (Cimpoi et al., 2014b) as OOD. For ImageNet, we use the same set of 20 classes chosen from ImageNet-22K as in (Hendrycks et al., 2019). These classes are chosen not to overlap with ImageNet-1k since the multi-label classifiers are pre-trained on ImageNet-1K. Specifically, we use the following classes for evaluating the MS-COCO and PASCAL-VOC pre-trained models: *dolphin, deer, bat, rhino, raccoon, octopus, giant clam, leech, venus flytrap, cherry tree, Japanese cherry blossoms, redwood, sunflower, croissant, stick cinnamon, cotton, rice, sugar cane, bamboo, and turmeric*. Since NUS-WIDE contains high-level concepts like animal, plants and flowers, we use a different set of classes that are distinct from NUS-WIDE: *asterism, battery, cave, cylinder, delta, fabric, filament, fire bell, hornet nest, kazoo, lichen, naval equipment, newspaper, paperclip, pythium, satellite, thumb, x-ray tube, yeast, zither*.

**Evaluation Metrics** We measure the following metrics that are commonly used for OOD detection: (1) the false positive rate (FPR95) of OOD examples when the true positive rate of in-distribution examples is at 95%; (2) the area under the receiver operating characteristic curve (AUROC); and (3) the area under the precision-recall curve (AUPR).

| $\mathcal{D}_{\text{in}}$ | MS-COCO | PASCAL-VOC | NUS-WIDE |
|---|---|---|---|
| | | FPR95 / AUROC / AUPR | |
| **OOD Score** | ↓ | ↑      ↑ | |
| **MaxLogit (Hendrycks et al., 2019)** | 43.53 / 89.11 / 93.74 | 45.06 / 89.22 / 83.14 | 56.46 / 83.58 / 94.32 |
| **MaxProb** | 43.53 / 89.11 / 93.74 | 45.06 / 89.22 / 83.14 | 56.46 / 83.58 / 94.32 |
| **MSP (Hendrycks & Gimpel, 2016)** | 79.90 / 73.70 / 85.37 | 74.05 / 79.32 / 72.54 | 88.50 / 60.81 / 87.00 |
| **ODIN (Liang et al., 2018)** | 43.53 / 89.11 / 93.74 | 45.06 / 89.22 / 83.16 | 56.46 / 83.58 / 94.32 |
| **Mahalanobis (Lee et al., 2018)** | 46.86 / 88.59 / 93.85 | 41.74 / 88.65 / 81.12 | 62.67 / 84.02 / 95.25 |
| **LOF (Breunig et al., 2000)** | 80.44 / 73.95 / 86.01 | 86.34 / 69.21 / 58.93 | 85.21 / 67.75 / 89.61 |
| **Isolation Forest (Liu et al., 2008)** | 94.39 / 49.04 / 66.87 | 93.22 / 50.67 / 35.78 | 95.69 / 53.12 / 83.32 |
| **MaxEnergy** | 43.53 / 89.11 / 93.74 | 45.06 / 89.22 / 83.14 | 56.46 / 83.58 / 94.32 |
| **SumEnergy** | **33.48 / 92.70 / 96.25** | **41.01 / 91.10 / 86.33** | **48.98 / 88.30 / 96.40** |

Table 1: OOD detection performance comparison using energy-based approaches vs. competitive baselines. We use DenseNet (Huang et al., 2017) to train on the in-distribution datasets. We use a subset of ImageNet classes as OOD test data, as described in Section 3.1. All values are percentages. ↑ indicates larger values are better, and ↓ indicates smaller values are better. **Bold** numbers are superior results. Description of baseline methods, additional evaluation results on different OOD test data, and different architecture (*e.g.*, ResNet) can be found in the Appendix.

## 3.2 RESULTS

**How do energy-based approaches compare to common OOD detection methods?** In Table 1, we compare energy-based approaches against competitive OOD detection methods in literature, where *SumEnergy* demonstrates state-of-the-art performance. For fair comparisons, we consider approaches that rely on pre-trained models (without performing retraining or fine-tuning). Following the set up in (Hendrycks et al., 2019), all the numbers reported are evaluated on ImageNet OOD test data, as described in Section 3.1. We provide additional evaluation results for the Texture OOD test dataset in Appendix A. Most baselines such as MaxLogit (Hendrycks et al., 2019), Maximum Softmax Probability (MSP) (Hendrycks & Gimpel, 2016), ODIN (Liang et al., 2018) and Mahalanobis (Lee et al., 2018) derive OOD indicator scores based on the maximum-valued statistics among all labels. Local Outlier Factor (LOF) (Breunig et al., 2000) uses K-nearest neighbors (KNN) to estimate the local density, where OOD examples are detected from having lower density compared to their neighbors. Isolation forest (Liu et al., 2008) is a tree-based approach, which detects anomaly based on the path length from the root node to the terminating node.

Among different approaches, *SumEnergy* outperforms the best-performing baseline across all three multi-label classifiers considered. In particular, on a network trained with the MS-COCO dataset, *SumEnergy* reduces FPR95 by **10.05**%, compared to MaxLogit. We provide the AUROC curves for our method SumEnergy in Figure 2, for all three in-distribution datasets considered. The y-axis is the true positive rate (TPR), whereas the x-axis is the FPR. The curves indicate how the OOD detection performance changes as we vary the threshold $\tau$ in Equation 17. We additionally evaluate on a different architecture, ResNet (He et al., 2016), for which we observe consistent improvement and provide details in the Appendix.

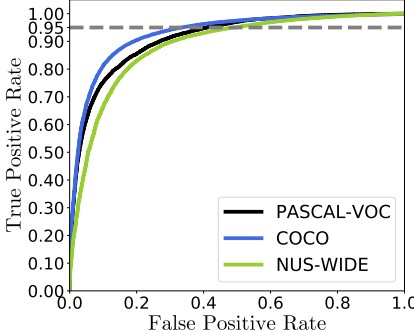

Figure 2: AUROC curves for OOD detector obtained from three in-distribution multi-label classification datasets.

We also note here that existing approaches using a pre-trained model (such as ODIN (Liang et al., 2018) and Mahalanobis (Lee et al., 2018)) have hyperparameters that need to be tuned. In contrast, using an energy-based method on pre-trained models is parameter-free and easy to use and deploy. In particular, the Mahalanobis approach is based on the assumption that feature representation forms class-conditional Gaussian distributions, and hence may not be well suited for the multi-label setting (which requires joint distribution to be learned).

**How do different aggregation methods affect OOD detection performance?** In Table 1, we also perform a comparative analysis of the effect of different aggregation functions that combine label-

| | $\mathcal{D}_{\text{in}}$ | MS-COCO | PASCAL | NUS-WIDE |
|---|---|---|---|---|
| | | **FPR95 / AUROC / AUPR** | | |
| **OOD Score** | **Aggregation** | ↓ ↑ ↑ | | |
| **Logit** | Sum | 95.46 / 61.81 / 80.39 | 87.18 / 72.68 / 61.24 | 96.53 / 51.75 / 82.55 |
| **Prob** | Sum | 45.04 / 89.32 / 94.40 | **38.57** / 86.53 / 79.10 | 50.84 / 83.82 / 95.15 |
| **ODIN** | Sum | 56.56 / 84.62 / 92.24 | 50.35 / 79.45 / 70.19 | 56.26 / 81.04 / 94.34 |
| **Mahalanobis** | Sum | 53.43 / 87.52 / 93.35 | 44.43 / 87.76 / 79.86 | 69.05 / 80.46 / 94.09 |
| **LOF** | Sum | N/A | N/A | N/A |
| **Isolation Forest** | Sum | N/A | N/A | N/A |
| **Energy (ours)** | Sum | **33.48 / 92.70 / 96.25** | 41.01 / **91.10 / 86.33** | 48.98 / **88.30 / 96.40** |

Table 2: Ablation study on the effect of summation for prior approaches. We use DenseNet (Huang et al., 2017) to train on the in-distribution datasets. We use ImageNet as OOD test data as described in Section 3.1. Note that *Sum* does not apply to tree-based or KNN-based approaches (e.g., LOF and Isolation Forest).

wise energy scores. Among those, we observe that *MaxEnergy* does not outperform *SumEnergy*, which utilizes information jointly from across the labels. The performance of *MaxEnergy* is on par with MaxLogit since *MaxEnergy*, given by $\max_i \log(1 + e^{f_{y_i}(\mathbf{x})})$, is approximately close to the MaxLogit when $f_{y_i}(\mathbf{x})$ is large. The results underline the importance of taking into account information from other labels, not just the maximum-valued label. This is because, in multi-label classification, the model may assign high probabilities to several classes. Theoretically, *SumEnergy* is also more meaningful, and can be interpreted from a joint likelihood perspective as shown in Section 2.2.

**What is the effect of applying the aggregation method to prior methods?** As an extension, we explore the effectiveness of applying the aggregation method to previous scoring functions such as logit (Hendrycks et al., 2019), MSP (Hendrycks & Gimpel, 2016) and ODIN (Liang et al., 2018). The results are summarized in Table 2. We calculate scores based on the logit $f_{y_i}(\mathbf{x})$, sigmoid of the logit $\frac{1}{1+e^{-f_{y_i}(\mathbf{x})}}$, ODIN score, as well as Mahalanobis distance score $M_{y_i}(\mathbf{x})$ by treating each label independently. We then perform summation across the label-wise scores as the overall OOD score. This ablation essentially replaces the *Max* aggregation with *Sum*, which helps understand the extent to which previous approaches are amenable in the multi-label setting. Note that the summation aggregation method does not apply to tree-based or KNN-based approaches such as LOF and Isolation Forest.

Interestingly, we found that applying summation over individual logit/MSP/ODIN/Mahalanobis scores from each label does not yield competitive results, and in many cases worsens the performance. For example, simply summing over the logits across the labels leads to severe degradation in performance since the outputs are mixed with positive and negative numbers. On MS-COCO, the FPR degrades from 43.53% using MaxLogit to 95.46% (using SumLogit). In contrast, *SumEnergy* does not suffer from this issue since the energy scores for each individual label have a uniform sign. More importantly, the label-wise scores derived from energy is theoretically more meaningful than logit/MSP/ODIN/Mahalanobis, since it is provably aligned with the probability density of the training data corresponding to the label. This underlines the importance of properly choosing the label-wise scoring function to be compatible with the aggregation method.

**SumEnergy vs. SumProb** We highlight the advantage of SumEnergy over SumProb both empirically and theoretically. As seen in Table 2, the performance difference between SumEnergy and SumProb is substantial. In particular, on MS-COCO, our method outperforms SumProb by 11.56% (FPR95). For threshold independent metric AUROC, SumEnergy consistently outperforms SumProb by 3.38% (MS-COCO), 4.57% (PASCAL), and 4.48% (NUS-WIDE). SumEnergy is a mathematically meaningful measurement and can be interpreted from a joint likelihood perspective (see Section 2.2), whereas SumProb does not. In fact, one can show that the probability score for each individual label is not aligned with the conditioned data density function. To see this, we can derive the probability for each binary logistic classifier as:

$$\log p(y_i \mid \mathbf{x}) = \log \frac{e^{f_{y_i}(\mathbf{x})}}{1 + e^{f_{y_i}(\mathbf{x})}} = f_{y_i}(\mathbf{x}) + E_{y_i}(\mathbf{x}).$$

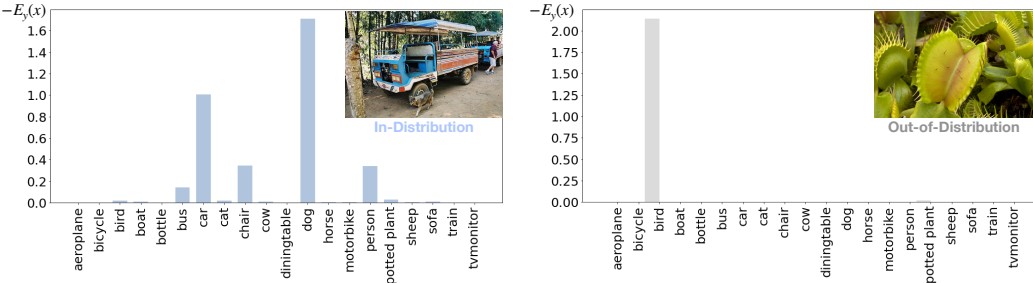

Figure 3: Label-wise energy scores $-E_{y_i}(\mathbf{x})$ for in-distribution example from PASCAL-VOC (left), and OOD input from ImageNet (right). The OOD input is misclassified using MaxLogit score since the dominant output has a high activation, making it indistinguishable from an in-distribution data's MaxLogit score. In contrast, *SumEnergy* correctly classifies both images since it results in larger differences in scores between in-distribution and OOD inputs.

In fact, the first term $f_{y_i}(\mathbf{x})$ is larger for in-distribution data with label $y_i$, whereas the second term is smaller for in-distribution data with $E_{y_i}(\mathbf{x}) \propto -\log p(\mathbf{x} \mid y_i)$. This leads to a biased scoring distribution that is no longer proportional to the label-conditional log-likelihood $\log p(\mathbf{x} \mid y_i)$:

$$\log p(y_i \mid \mathbf{x}) \not\propto \text{class-conditional likelihood of data with label } y_i$$

SumProb, as a result, inherits this weakness theoretically and performs worse than SumEnergy.

**Qualitative case study** Lastly, to provide further insights on our method, we qualitatively examine examples from the multi-label classification dataset PASCAL-VOC (in-dist.) and OOD input from the ImageNet that are correctly classified by *SumEnergy* but not MaxLogit. In Figure 3 (left), we see an in-distribution example is labeled as `dog`, `car`, `chair` and `person`, with *MaxLogit* score 1.63 and *SumEnergy* score 3.23. We also show an OOD input (Figure 3, right) with a single dominant activation on the `bird` class, with MaxLogit score 2.14 and *SumEnergy* score 2.19. In this example, taking the sum appropriately resulted in a higher score for the in-distribution image than the OOD image. Contrarily, MaxLogit's score for the in-distribution image was lower than that of the OOD image, which results in ineffective detection.

## 4 RELATED WORK

**Multi-label classification** The task of identifying multiple classes within an input example is of significant interest in many applications (Tsoumakas & KataKis, 2007), with deep neural networks being commonly used as the classifier. Natural images usually contain several objects and may have many associated tags (Wang et al., 2016). (Gong et al., 2013) use convolutional neural networks (CNN) to annotate images with 3 or 5 tags on the NUS-WIDE dataset. (Chen et al., 2019a) use CNNs to tag images of road scenes from 52 possible labels. In the medical domain, (Wang et al., 2017) present a chest X-ray dataset in which one image may contain multiple abnormalities. Multi-label classification is also prominent in natural language processing (Nam et al., 2014). Recent work also provides a theoretical analysis of multi-label classification under various measures (Wu & Zhu, 2020). Our proposed method is therefore relevant to a wide range of applications in the real world.

**Out-of-distribution uncertainty for pre-trained models** The softmax confidence score has become a common baseline for OOD detection (Hendrycks & Gimpel, 2016). A theoretical investigation (Hein et al., 2019) shows that neural networks with ReLU activation can produce arbitrarily high softmax confidence for OOD inputs. Several works attempt to improve the OOD uncertainty estimation by using deep ensembles (Lakshminarayanan et al., 2017), ODIN score (Liang et al., 2018), Mahalanobis distance-based confidence score (Lee et al., 2018), and generalized ODIN score (Hsu et al., 2020). (DeVries & Taylor, 2018) propose to learn the confidence score by using an auxiliary branch to derive the OOD score. Recent work using energy score demonstrated state-of-the-art performance on OOD detection tasks (Liu et al., 2020). However, previous methods primarily focused on multi-class classification networks. In contrast, in our work, we propose a parameter-free mea-

surement that allows effective OOD detection in the underexplored *multi-label* setting, where the information from across various labels are combined in a theoretically meaningful manner.

**Out-of-distribution detection with model fine-tuning** While our work primarily focused on OOD detection for pre-trained neural networks, a parallel line of research has also explored using auxiliary outlier OOD data to help the OOD detector generalize better. Auxiliary data allows the model to be explicitly regularized through fine-tuning, producing lower confidence on anomalous examples (Bevandić et al., 2018; Geifman & El-Yaniv, 2019; Malinin & Gales, 2018; Mohseni et al., 2020; Subramanya et al., 2017). A loss function is used to force the predictive distribution of OOD samples toward a uniform distribution (Lee et al., 2017). Recently, (Mohseni et al., 2020) explore training by adding a background class for an OOD score. (Chen et al., 2020) propose using hard outlier mining which improves the OOD detection performance on both clean and perturbed natural images. However, existing works have primarily focused on the multi-class classification setting. We leave the fine-tuning aspect with auxiliary data for multi-label classification as future exploration.

**Generative Modeling Based Out-of-distribution Detection.** Generative models (Dinh et al., 2016; Kingma & Welling, 2013; Rezende et al., 2014; Van den Oord et al., 2016; Tabak & Turner, 2013) can be alternative approaches for detecting OOD examples, as they directly estimate the in-distribution density and can declare a test sample to be out-of-distribution if it lies in the low-density regions. However, as shown by (Nalisnick et al., 2018), deep generative models can assign a high likelihood to out-of-distribution data. Deep generative models can be more effective for out-of-distribution detection using improved metrics (Choi & Jang, 2018), including likelihood ratio (Ren et al., 2019; Serrà et al., 2019). Though our work is based on discriminative classification models, we show that label-wise energy scores can be theoretically interpreted from a data density perspective. More importantly, generative based models (Hinz et al., 2019) can be prohibitively challenging to train and optimize, especially on large and complex multi-label datasets that we considered (*e.g.*, MS-COCO, NUS-WIDE etc). In contrast, our method relies on a discriminative multi-label classifier, which can be much easier to optimize using standard SGD.

**Energy-based learning** Energy-based machine learning models date back to Boltzmann machines (Ackley et al., 1985; Salakhutdinov & Larochelle, 2010). Energy-based learning (LeCun et al., 2006; Ranzato et al., 2007a;b) provides a unified framework for many probabilistic and non-probabilistic approaches to learning. Recent work (Zhao et al., 2019) also demonstrated using energy functions to train GANs (Goodfellow et al., 2014), where the discriminator uses energy values to differentiate between real and generated images. Xie et al. (2016) first showed that a discriminative classifier can be interpreted from an energy-based perspective. Subsequent works (Xie et al., 2017; 2019; 2018b;a) explored video generation, 3D shape pattern generation, and text generation (Deng et al., 2019) through EBMs. Energy-based methods are also used in structure prediction (Belanger & McCallum, 2016; Tu & Gimpel, 2018). Grathwohl et al. (2019) showed that a discriminative classifier can be interpreted from an energy-based perspective. The proposed JEM optimization objective estimates the joint distribution $p(\mathbf{x}, y)$ from a generative perspective, which requires estimating the normalized densities and can be intractable and unstable to compute. Liu et al. (2020) propose to use energy score for OOD detection that is derived from a pure discriminatively trained classifier, which demonstrated superior performance for multi-class classification networks. In contrast, our work focuses on a multi-label setting, where we contribute aggregation methods that utilize information jointly from across all labels.

## 5 CONCLUSION

In this work, we propose energy scores for OOD detection in the multi-label classification setting. We show that aggregating energies over all labels into *SumEnergy* results in better discrimination between in-distribution and OOD inputs compared to using information from only one label's information (i.e. MaxLogit, MSP, ODIN, or Mahalanobis). Additionally, we justify the mathematical interpretation of *SumEnergy* from a joint likelihood perspective. *SumEnergy* obtains better OOD detection performance compared to competitive baseline methods, establishing new state-of-the-art on this task. Applications of multi-label classification can benefit from our methods, and we anticipate further research in OOD detection to extend this work. We hope our work will increase the attention toward a broader view of OOD uncertainty estimation from an energy-based perspective.

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

# A APPENDIX

## A.1 EVALUATION ON DIFFERENT ARCHITECTURE

We provide additional evaluation results for ResNet (He et al., 2016). The classifiers have a ResNet-101 backbone architecture, but with a final layer that is replaced by 2 fully connected layers. Each classifier is pre-trained on ImageNet-1K and then fine-tuned with the logistic sigmoid function to its corresponding multi-label dataset. We use the same training settings as in the main paper. After training, the mAP is 87.73% for PASCAL-VOC, 72.77% for MS-COCO, and 61.47% for NUS-WIDE.

In Table 3, we show the performance comparison of various OOD detection approaches, evaluated on ImageNet as the OOD test set. The ablation of applying summation over baseline methods is provided in Table 4.

| OOD Score | Aggregation | MS-COCO
FPR95 / AUROC / AUPR
↓ ↑ ↑ | PASCAL-VOC | NUS-WIDE |
|---|---|---|---|---|
| Logit | Max | 34.54 / 90.93 / 94.30 | 36.32 / 91.04 / 82.68 | 58.05 / 83.07 / 94.21 |
| Prob | Max | 34.54 / 90.93 / 94.30 | 36.32 / 91.04 / 82.68 | 58.05 / 83.07 / 94.21 |
| MSP | Max | 77.92 / 72.43 / 84.34 | 69.85 / 78.24 / 67.93 | 88.75 / 59.19 / 86.40 |
| ODIN | Max | 34.58 / 90.26 / 93.69 | 36.32 / 91.04 / 82.68 | 58.05 / 83.07 / 94.21 |
| Mahalanobis | Max | 94.04 / 49.49 / 70.71 | 78.02 / 70.93 / 59.84 | 61.33 / 83.75 / 95.15 |
| LOF | - | 74.30 / 74.87 / 85.82 | 76.71 / 67.54 / 55.35 | 85.42 / 69.37 / 90.36 |
| Isolation Forest | - | 99.06 / 37.59 / 63.43 | 98.64 / 41.94 / 33.50 | 96.59 / 50.75 / 82.91 |
| Energy | Max | 34.54 / 90.93 / 94.30 | 36.32 / 91.04 / 82.68 | 58.05 / 83.07 / 94.21 |
| (ours) | Sum | **31.51 / 92.68 / 96.15** | **31.96 / 92.32 / 86.87** | **50.25 / 88.12 / 96.34** |

Table 3: OOD detection performance comparison using energy-based approaches vs. competitive baselines. We use ResNet (He et al., 2016) to train on the in-distribution datasets. We use a subset of ImageNet classes as OOD test data, as described in Section 3.1. All values are percentages. ↑ indicates larger values are better, and ↓ indicates smaller values are better. **Bold** numbers are superior results.

| OOD Score | Aggregation | MS-COCO
FPR95 / AUROC / AUPR
↓ ↑ ↑ | PASCAL | NUS-WIDE |
|---|---|---|---|---|
| Logit | Sum | 95.63 / 53.52 / 73.25 | 96.36 / 49.44 / 43.07 | 96.49 / 49.83 / 81.78 |
| Prob | Sum | 43.69 / 87.21 / 93.14 | 35.97 / 84.68 / 76.61 | 55.86 / 82.97 / 94.92 |
| ODIN | Sum | 43.69 / 87.21 / 93.14 | 53.77 / 74.50 / 67.15 | 55.24 / 81.84 / 94.59 |
| Mahalanobis | Sum | 94.47 / 46.82 / 67.06 | 78.56 / 70.84 / 59.34 | 62.79 / 83.19 / 94.96 |
| LOF | Sum | N/A | N/A | N/A |
| Isolation Forest | Sum | N/A | N/A | N/A |
| Energy | Sum | **31.51 / 92.68 / 96.15** | **31.96 / 92.32 / 86.87** | **50.25 / 88.12 / 96.34** |

Table 4: Ablation study on the effect of aggregation methods for prior approaches. We use ResNet (He et al., 2016) to train on the in-distribution datasets. We use ImageNet as OOD test data as described in Section 3.1. Note that *Sum* is not applicable to tree-based or KNN-based approaches (e.g., LOF and Isolation Forest).

## A.2 EVALUATION ON DIFFERENT OOD TEST DATA

In addition to ImageNet, we also evaluate on a different OOD test dataset, Textures (Cimpoi et al., 2014a). The results are reported in Table 5 and Table 6.

## A.3 BASELINE METHODS

In multi-label classification, the prediction for each label $y_i$ with $i \in \{1, 2, ..., K\}$ is independently made by a binary logistic classifier:

$$p(y_i \mid \mathbf{x}) = \frac{e^{f_{y_i}(\mathbf{x})}}{1 + e^{f_{y_i}(\mathbf{x})}}.$$

| | $\mathcal{D}_{\text{in}}$ | MS-COCO | PASCAL | NUS-WIDE |
|---|---|---|---|---|
| | | **FPR95 / AUROC / AUPR** | | |
| **OOD Score** | **Aggregation** | ↓  ↑  ↑ | | |
| **Logit** | Max | 14.63 / 96.10 / 99.32 | 12.36 / 96.22 / 96.97 | 38.46 / 87.42 / 97.19 |
| **Prob** | Max | 14.63 / 96.10 / 99.32 | 12.36 / 96.22 / 96.97 | 38.46 / 87.42 / 97.19 |
| **MSP** | Max | 60.82 / 83.70 / 97.05 | 41.81 / 89.76 / 93.00 | 83.09 / 63.41 / 92.48 |
| **ODIN** | Max | **12.22** / 96.18 / 99.29 | 12.36 / 96.22 / 96.97 | 38.46 / 87.42 / 97.19 |
| **Mahalanobis** | Max | 44.61 / 85.71 / 97.41 | 19.17 / 96.23 / **97.90** | 36.19 / 91.36 / 98.52 |
| **LOF** | - | 70.16 / 74.73 / 94.96 | 89.49 / 60.37 / 76.70 | 64.27 / 78.23 / 95.94 |
| **Isolation Forest** | - | 95.55 / 53.21 / 90.45 | 99.59 / 20.89 / 50.11 | 93.07 / 51.01 / 89.17 |
| **Energy** | Max | 14.63 / 96.10 / 99.32 | 12.36 / 96.22 / 96.97 | 38.46 / 87.42 / 97.19 |
| **(ours)** | Sum | 12.82 / **96.84 / 99.54** | **10.87 / 96.78** / 97.87 | **31.68 / 92.43 / 98.65** |

Table 5: Texture as OOD data. We use ResNet (He et al., 2016) to train on the in-distribution datasets. All values are percentages. ↑ indicates larger values are better, and ↓ indicates smaller values are better. **Bold** numbers are superior results.

| | $\mathcal{D}_{\text{in}}$ | MS-COCO | PASCAL | NUS-WIDE |
|---|---|---|---|---|
| | | **FPR95 / AUROC / AUPR** | | |
| **OOD Score** | **Aggregation** | ↓  ↑  ↑ | | |
| **Logit** | Sum | 95.63 / 53.52 / 73.25 | 96.36 / 49.44 / 43.07 | 92.38 / 52.72 / 89.21 |
| **Prob** | Sum | 43.69 / 87.21 / 93.14 | 35.97 / 84.68 / 76.61 | 34.88 / 90.76 / 98.57 |
| **ODIN** | Sum | 43.69 / 87.21 / 93.14 | 53.77 / 74.50 / 67.15 | 35.27 / 89.36 / 98.31 |
| **Mahalanobis** | Sum | 45.62 / 84.34 / 97.02 | 19.45 / 96.09 / 97.80 | 37.55 / 91.04 / 98.47 |
| **LOF** | Sum | N/A | N/A | N/A |
| **Isolation Forest** | Sum | N/A | N/A | N/A |
| **Energy** | Sum | **12.82 / 96.84 / 99.54** | **10.87 / 96.78 / 97.87** | **31.68 / 92.43 / 98.65** |

Table 6: Ablation study on the effect of aggregation methods for prior approaches. We use ResNet (He et al., 2016) to train on the in-distribution datasets. We use Texture (Cimpoi et al., 2014a) as OOD test data as described in Section 3.1. Note that *Sum* is not applicable to tree-based or KNN-based approaches (e.g., LOF and Isolation Forest).

We consider the following baselines methods under *maximum* aggregation:

$$\textbf{MaxLogit} = \max_i f_{y_i}(\mathbf{x}) \tag{18}$$

$$\textbf{MaxProb} = \max_i p(y_i \mid \mathbf{x}) = \max_i \frac{e^{f_{y_i}(\mathbf{x})}}{1 + e^{f_{y_i}(\mathbf{x})}} \tag{19}$$

$$\textbf{MSP} = \max_i \frac{e^{f_{y_i}(\mathbf{x})}}{\sum_j^K e^{f_{y_j}(\mathbf{x})}} \tag{20}$$

$$\textbf{ODIN} = \max_i \frac{e^{f_{y_i}(\hat{\mathbf{x}})/T}}{1 + e^{f_{y_i}(\hat{\mathbf{x}})/T}} \tag{21}$$

$$\textbf{Mahalanobis} = \max_i -(\phi(\hat{\mathbf{x}}) - \hat{\mu}_{y_i})^\mathsf{T} \hat{\Sigma}^{-1} (\phi(\hat{\mathbf{x}}) - \hat{\mu}_{y_i}) \tag{22}$$

In particular, ODIN was originally designed for multi-class but we adapt for the multi-label case by taking the maximum of calibrated label-wise predictions. The input perturbation is calculated using $\hat{\mathbf{x}} = \mathbf{x} - \epsilon \text{sign}(-\nabla \ell_{\hat{y}_i})$, where $\ell_{\hat{y}_i}$ is the binary cross-entropy loss for the label $\hat{y}_i$ with the largest output, i.e., $\hat{y}_i = \arg\max_i p(y_i = 1 \mid \mathbf{x})$. For Mahalanobis distance, we extract the feature embedding $\phi(\mathbf{x})$ for a given sample. $\hat{\mu}_{y_i}$ is the class conditional mean for label $y_i$, and $\hat{\Sigma}^{-1}$ is the covariant matrix.

## A.4 VALIDATION DATA FOR BASELINES

We use a combination of the following validation datasets to select hyperparameters for ODIN (Liang et al., 2018) and Mahalanobis (Lee et al., 2018). The validation set consists of:

- Gaussian noise sampled i.i.d. from an isotropic Gaussian distribution;
- uniform noise where each pixel is sampled from $U = [-1, 1]$;
- In-distribution data corrupted into OOD data by applying (1) pixel-wise arithmetic mean of random pair of in-distribution images; (2) geometric mean of random pair of in-distribution images; and (3) randomly permuting 16 equally sized patches of an in-distribution image.

## A.5 Hyperparameter tuning for baselines

ODIN (Liang et al., 2018) and Mahalanobis (Lee et al., 2018) require hyper-parameter tuning, such as temperature and magnitude of noise $\epsilon$. We use the validation data above for selecting the optimal hyperparameters. For ODIN, temperature T is chosen from [1,10,100,1000] and the perturbation magnitude $\epsilon$ is chosen from 21 evenly spaced numbers starting from 0 and ending at 0.004. For Mahalanobis, the perturbation magnitude $\epsilon$ is chosen from [0, 0.0005, 0.0014, 0.001, 0.002, 0.005]. The optimal parameters are chosen to minimize the FPR at TPR95 on the validation set.

