# OpenReview forum: "Energy-based Out-of-distribution Detection for Multi-label Classification"
_ICLR.cc/2021/Conference — Reject_

### Official Review · AnonReviewer3 · 2020-10-28
**clear work, however, the contribution could a little limited.**

**Rating:** 6
**Confidence:** 3

**Review:**

Summary: In this work, SumEnergy is proposed for out-of-distribution detection for multi-label classification. According to the results of the experiment, SumEnergy performs better than MaxEnergy in several datasets of multi-label classification.

+ves:
1. This paper is written well. The main part is clear.  With the mathematical equation, it shows that SumEnergy is linearly aligned with the log of joint likelihood. The aggregated include more likelihood information over all labels.

2. The results in Tables 1 and 2 show a clear comparison with other baselines. It also indicates that SumEnergy performs best among all methods. The analysis and Qualitative case study are convincing.

Concerns
1. Most baselines are designed by the authors. Unfortunately, there are no other baselines that are done by other papers. The contribution of this work could be a little questionable.

2. The SumEnergy seems to has a minor difference with Sum Prob.  Maybe more reason on why still use energy score could be discussed.

Questions during the rebuttal period:


Some related work on energy-based learning could be mentioned.:

[1] Structured Prediction Energy Networks, ICML 2016

[2] Learning Approximate Inference Networks for Structured Prediction. ICLR 2018

[3] Residual Energy-Based Models for Text Generation. ICLR 2020

---

> ### Author Response · Authors · 2020-11-12
> **[draft updated] Thank you for the positive feedback!**
>
> We thank the reviewer for the positive feedback. We are glad to hear that our paper is well written, with convincing and strong results that outperform baseline methods. Moreover, we are pleased that R3 appreciated our mathematical insights on why SumEnergy is well-suited for the multi-label classification problem.
>
> 1 Re: Most baselines are designed by authors.
>
> We’d like to clarify that MaxLogit, LOF, and IForest are existing methods, which were evaluated in [1] for the OOD detection in multi-label classification networks. Our evaluation framework builds on [1], while expanding with more competitive baselines in literature. We are not inventing things new here and would like to give credit to prior works. **We’ve updated our draft, added clear citations to each baseline method in Table 1**.
>
> [1] Dan Hendrycks, Steven Basart, Mantas Mazeika, Mohammadreza Mostajabi, Jacob Steinhardt, and
> Dawn Song. A benchmark for anomaly segmentation, 2019.
>
>
> 2 Re: SumEnergy vs SumProb
>
> We highlight the advantage of SumEnergy over SumProb both empirically and theoretically. **See expanded discussion on Page 7-8**.
>
> **Empirically**: The performance difference between SumEnergy and SumProb is substantial, as evidenced in Table 2. In particular, on MS-COCO, our method outperforms SumProb by 11.56% (FPR95). For threshold independent metric AUROC, SumEnergy consistently outperforms SumProb by 3.38% (MS-COCO), 4.57% (PASCAL), and 4.48% (NUS-WIDE).
>
> **Theoretically**: SumEnergy is a mathematically meaningful measurement and can be interpreted from a joint likelihood perspective (see Section 2.2), whereas SumProb does not. In fact, one can show that the probability score for each individual label is not aligned with the conditioned data density function. To see this, we can derive the probability for each binary logistic classifier as:
> \begin{align*}
> 	\log p(y \mid \textbf{x}) &=  \log \frac{e^{f_y(\textbf{x})}}{1+ e^{f_{y}(\textbf{x})}} = f_y(\textbf{x}) + E_y(\textbf{x}).
> \end{align*}
> In fact, the first term $f_y(\textbf{x})$ is larger for in-distribution data with label $y$, whereas the second term is smaller for in-distribution data with $E_y(\textbf{x}) \propto -\log p(\textbf{x}\mid y)$. This leads to a biased scoring distribution that is no longer proportional to the label-conditional likelihood $\log p(\textbf{x}\mid y)$. In other words:
> \begin{align*}
> 	\log p(y \mid \textbf{x}) \not \propto \text{data density for label}~y
> \end{align*}
>
> SumProb, as a result, inherits this weakness theoretically and performs worse than SumEnergy.
>
> 3 Re: Missing references for energy-based learning
> Thank you - all added in our new draft (**see Page 8-9**)!
>
> In summary, we believe our work contributes to the field by studying an **underexplored problem** (OOD detection for multi-label classification), proposing an **unexplored and effective solution space** (OOD scoring function estimated jointly from multiple labels, with up to 10.05% FPR reduction compared to the previous best method) that establishes state-of-the-art performance. We also provide theoretical interpretations. Furthermore, our work opens up an **interesting and promising** future direction for OOD detection to consider information jointly from across semantics.
>
> We thank the reviewer again for the constructive comments which helped greatly improve our work!

---

### Official Review · AnonReviewer1 · 2020-10-28
**The explanation is confusing**

**Rating:** 4
**Confidence:** 4

**Review:**

This paper deals with the out-of-distribution detection task in the multi-label classification (MLC) setting. It proposes an energy-based method called SumEnergy, which estimates the OOD indicator scores by aggregating energy scores from multiple labels. Experimental results illustrate its superiority compared with other baselines.


##############################################################################################
pros:
1. Overall, this paper is well-written and organized.
2. The proposed method achieves promising experimental results.

##############################################################################################
cons:
1. The explanation for why SumEnergy works is confusing. Specifically,  in Eq.(13), authors explain $E_{sum}(\mathcal{x})$ is linearly aligned with the log of joint likelihood, which is confusing, because $p(\mathcal{x}, \mathcal{y}) = p(\mathcal{x}, y_1) p( y_2 \| \mathcal{x}, y_1)  ...  p( y_K \| \mathcal{x}, y_1, ... , y_{K-1})$ while Eq.(13) has $\prod_{j=1}^{K} p(\mathcal{x}, y_j)$. (For clarity, I have changed some notations.)
2. What is the effect of the hyperparameter $\tau$ for these two methods (i.e. MaxEnergy and SumEnergy)? Please give more comparisons and clarify how to tune $\tau$.
3. The novelty is incremental. Although authors highlight the non-triviality to extend from the multi-class setting (Liu et al., 2020) to multi-label setting lies in leveraging information between different labels,  the proposed method SumEnergy may not reflect this point due to 1.

---

> ### Author Response · Authors · 2020-11-12
> **[draft updated] clarification on math explanation, threshold, and novelty. Thank you for the insightful comments!**
>
> We thank the reviewer for the insightful and helpful comments! We are glad to hear that our paper is well written with superior and promising results. Below we address all 3 comments in detail.
>
> 1. Mathematical explanation:
>
> To clarify this, the interpretation from a joint likelihood can be seen from Equation 11, where $E_\text{sum}(\textbf{x}) = \sum_{y=1}^K \log p(\textbf{x}|y) + Z = \log \prod_{y=1}^K p(\textbf{x}| y) + Z$. In this equation, $\prod_{y=1}^K p(\textbf{x}| y)$ measures the joint conditioned likelihood for the sample x to be in each class. Equation 13 shows a more direct meaning behind our method, which is equivalent to the probability for the data to be associated with all labels. Intuitively, OOD data should have a low SumEnergy score since $p(\textbf{x},y)$ will be low for all labels.
>
> Our derivation is based on the assumption that all labels are conditionally independent, in which case reduces the chain probability to $\prod_{y=1}^K p(\textbf{x},y)$. This assumption is consistent with standard multi-label classification loss, which is multiple independent binary classification problems. We consider this setting for its simplicity and generality. Our work also opens up an interesting future direction of OOD detection by considering the structural dependency among labels. **We have further clarified this in our updated draft (see Page 4, Section 2.2)**.
>
> 2. Threshold:
>
> We follow the common practice in literature and select $\tau$ based on 95% of true positive rate (fraction of in-distribution images that are correctly classified). Performance reported in Table 1 shows the average FPR when TPR is 95%. **To see the sensitivity of the threshold, we additionally provide AUROC curves in Figure 2**. This shows how the FPR changes (in x-axis) as we sweep over all thresholds $\tau$ (which translates into different TPR accordingly). In general, our method produces high AUROC consistently across three in-distribution datasets considered.
>
>
> 3. Novelty:
>
> While the energy score was used for OOD detection in the multi-class setting (Liu et al., 2020), this method does not directly generalize to a multi-label classification setting. A multi-label setting in fact exacerbates the difficulty of OOD detection since labels are not mutually exclusive. The key technical challenge and contribution of our work are to derive statistics by considering the joint information across labels (which is indeed reflected in our formulation as clarified above). This joint estimation of the OOD scoring function was previously unexplored in literature, hence not incremental but new to our work.
>
> In particular, we showed in Section 2.2 the novel definition of SumEnergy derived from a multi-label classification network, as well as its mathematical interpretations from a joint likelihood view---both of which are new contributions to this work, and are distinct from prior works.
>
> Our empirical results in Table 1 support precisely the importance of taking into account joint information from across labels, which was not considered in previous baselines.
>
> In summary, our work contributes to the field by both studying an **underexplored problem** (OOD detection for multi-label classification) and proposing an **unexplored solution space** (OOD scoring function estimated jointly from multiple labels). Furthermore, it opens up an **interesting and promising** future direction for OOD detection to consider information jointly from across semantics.

---

> > ### Comment · AnonReviewer1 · 2020-11-19
> > **The math explanation is still confusing**
> >
> > Thanks for your response.  However, the mathematical explanation is still confusing.
> >
> > For the math explanation part:
> >
> > Firstly, the mathematical symbols are confusing. In Eq.(12), $p(\mathbf{x}, y_j)$ (j = 1,...,K) exists while $p(\mathbf{x} | y)$ exists in other equations (although I know what the authors aim to mean). Actually, $p(\mathbf{x} | y = k)$ means the conditional probability $p(\mathbf{x} | y_k = 1)$. The symbols should be unified and given clear meaning descriptions.
> >
> > Secondly, the authors clarify that the interpretation from a joint likelihood can be seen from Eq.(11), where $E_{sum} = \sum_{i=1}^{K} \log p(\mathbf{x} | y) + Z = \log \prod_{j=1}^K p(\mathbf{x} | y) + Z$. Authors claim that $\prod_{j=1}^K p(\mathbf{x} | y)$ measures the joint conditioned likelihood for the sample $x$ to be in each class, while the joint conditional likelihood should be $p(\mathbf{x} | y_1 = 1, y_2 = 1, ... , y_K = 1)$.
> >
> > I concern the authors don't express precisely. Although the authors claim that the deviations are based on the assumption that all labels are independent, in my opinion, in this case, it cannot reduce the chain probability to $\prod_{j=1}^K p(\mathbf{x}, y)$ (i.e. $\prod_{j=1}^K p(\mathbf{x}, y_j = 1)$) while its original probability should be $p(\mathbf{x}, y_1 = 1, y_2 = 1, ... , y_K = 1)$. Please give a formal proof if you could. Besides, I agree that SumEnergy can aggregate more information over the labels than MaxEnergy which only considers the largest label-wise energy score among all labels. Furthermore, recent theory [1] has confirmed the superiority of the standard multi-label classification loss for many measures (not only for Hamming Loss, but also for Subset Accuracy in small label space case).
> >
> > [1] Multi-label classification: do Hamming loss and subset accuracy really conflict with each other?, NeurIPS 2020

---

> > > ### Author Response · Authors · 2020-11-20
> > > **We have addressed all the concerns and believe the math is clear; please see the updated draft for formal proof**
> > >
> > > Thanks for the response. We are glad to hear that **the reviewer agrees on the significance of our proposed SumEnergy**, which aggregates information over the labels and is better than methods that only consider the maximum scores. We have addressed the concerns regarding math clarity.
> > >
> > > 1. [notation] As per suggestion, we’ve updated the notations throughout the paper for clarity, using $y_i$ to indicate when an input $\textbf{x}$ is associated with the $i$-th label. **See changes made in Equations (1) - (16) in Section 2,  as well as notations in Section 3, Page 7-8**.
> > >
> > > 2. We expanded our draft by adding formal derivations in forms of joint conditional likelihood  $p(\textbf{x}\mid y_1=1,y_2=1,...,y_K=1)$. **See detailed proof in Section 2.2, Equation (12-16) on Page 4**.  To provide a brief proof here for your convenience, we can apply the Bayes' theorem in each term of $\log p(\textbf{x} \mid y_i)$ in $E_\text{sum}$, which yields:
> > >
> > > \begin{align}
> > >    E_\text{sum}(\textbf{x})&= \log \prod_{i=1}^K  \frac{p(y_i \mid \textbf{x}) \cdot p(\textbf{x})}{p(y_i)} + Z\\
> > >     = \log \prod_{i=1}^K p(y_i \mid \textbf{x})  + K\cdot \log p(\textbf{x}) + \underbrace{(Z - \log \prod_{i=1}^K p(y_i))}_{\text{C: constant for all  $\textbf{x}$}}
> > > \end{align}
> > >
> > > Given all label $y_i$ are conditionally independent given $\textbf{x}$, we have $\prod_{i=1}^K p(y_i \mid \textbf{x}) = p(y_1,y_2,...,y_K\mid \textbf{x})$. Therefore, Equation above is equivalent to:
> > >
> > > $$  E_\text{sum}(\textbf{x})= \log p(y_1,y_2, \ldots, y_K \mid \textbf{x}) + K\cdot \log p(\textbf{x}) + C$$
> > > $$= \log \frac{p(\textbf{x} \mid y_1,y_2,...,y_K)\cdot \prod_{i=1}^K p(y_i)}{p(\textbf{x})} + K\cdot \log p(\textbf{x})  + C\text{
> > >      (Note: first term is via the Bayes' theorem)}$$
> > > $$=\underbrace{\log p(\textbf{x} \mid y_1, y_2,...,y_K)}_{\text{joint conditional log likelihood}}+ (K-1) \cdot \log p(\textbf{x}) + Z$$
> > >
> > > Our proposed SumEnergy therefore captures the underlying data density and the joint information across labels, which is desirable for OOD detection in a multi-label setting. The first term is precisely joint conditional log likelihood, and also highlights the novelty and importance to our work (which is to take into account the joint estimation in a multi-label setting). This also differentiates our method from the multi-class setting which does not consider this term otherwise.
> > >
> > > 3. Reference to Wu et al. has been added. See Page 8, related work.
> > >
> > > In summary, **we greatly appreciate the feedback and have sufficiently addressed all the concerns of the reviewer**. We hope the reviewer can evaluate our work by considering the importance of the research problem (OOD detection in multi-label setting), technical solution (OOD score estimation jointly from multiple labels through energy), as well as the significance of the results (better by a large margin, with up to 10.05% FPR reduction over current SOTA).

---

> ### Author Response · Authors · 2020-11-18
> **follow up**
>
> Dear reviewer,
>
> We believe all three comments raised have been addressed in our response and updated draft. We'd like to follow up and clarify any remainder confusion.
>
> Your feedback has been very important for us to improve the work!
>
> Thank you,
> Authors of Paper1862

---

### Official Review · AnonReviewer2 · 2020-10-28
**The paper proposes a SumEnergy method to estimate the out-of-distribution indicator scores for multi-label classification.**

**Rating:** 6
**Confidence:** 3

**Review:**

####################

Summary:

The paper proposes a SumEnergy method to estimate the out-of-distribution indicator scores for multi-label classification.

####################

Reason for score:

Overall, the paper is above the borderline. I like the idea of utilizing SumEnergy operation to address the out-of-distribution problem in the task of multi-label classification. My major concern is about some unclear parts in the paper and insufficient experimental comparison (see cons below). Hopefully, it would be grateful that the authors could address my concerns during the rebuttal period.


####################

Pros:

(1) The motivation of the paper, i.e., out-of-distribution detection in multi-label classification is very important and deserves research further.

(2) Extensive experimental results demonstrate that the SumEnergy based on the aggregation over label-wise energy scores achieves better performance than that of MaxEnergy.

(3) The comparative analysis and ablation study in the paper are convinced and detailed, which can help better understand the applicability in the multi-label setting.

####################

Cons:

(1) Even though the experimental results demonstrate that the proposed SumEnergy outperforms MaxEnergy and MaxLogit, it will be better to conduct additional comparisons with other methods that also consider the information of all the labels, such as attention-based methods [1][2] and graph-based [3][4] multi-label classification approaches.

[1] Recurrent attentional reinforcement learning for multi-label image recognition. AAAI 2018
[2] Decoupling category-wise independence and relevance with self-attention for multi-label image classification. Arxiv 2019
[3] Learning semantic-specific graph representation for multi-label image recognition. Arxiv 2019
[4] Multi-label image recognition with graph convolutional networks. CVPR 2019

(2) According to the description in the paper, the OOD problem is similar to the few-shot or zero-shot issue. Could the authors explain more about the difference between OOD and few-shot/zero-shot learning, please? Moreover, more experimental details and results on few-shot/zero-shot problems will be welcome.

(3) How to determine the energy threshold seems very critical to the task, because the scores always various on different dataset and domains with the proposed SumEnergy method. More discussion and the impact for experimental results should be added in the paper.

####################
Questions during the rebuttal period:

Please address and clarify the cons above. Thank you!

---

> ### Author Response · Authors · 2020-11-12
> **[draft updated] Thank you for the constructive feedback!**
>
> We thank the reviewer for the constructive feedback! We are encouraged to hear that our motivation and idea to be appealing and that our analysis and evaluations are extensive and strong. We absolutely agree with your view that OOD detection in multi-label classification deserves research further. Below we address the comments in detail.
>
> Re: Effect of energy threshold $\tau$
>
> Great question! We follow the common practice in literature and select $\tau$ based on 95% of true positive rate (fraction of in-distribution images that are correctly classified). To see the effect of $\tau$, **we have updated the draft with the AUROC curves in Figure 2, Page 6**. The curves show how the performance changes (FPR, x-axis) as we vary the TPR (that determines $\tau$) on the y-axis. We report AUROC performance in Table 1, which is a threshold-independent measurement for OOD detection. In general, our method produces high AUROC consistently across three in-distribution datasets considered.
>
> Re: Attention-based & graph-based methods for multi-label classification
>
> Very interesting suggestions! Our main focus in the paper is to derive OOD scoring statistics from the output of a pre-trained multi-label classification model, rather than improving the multi-label classification model itself. We opt for the standard multi-label classification network for its generality and simplicity. However, we concur that improved training mechanisms (including suggested graph-based & attention-based methods) for the classification network can be beneficial for improving the downstream OOD scoring estimation. **We have highlighted these works on Page 4 in Section 2.2**, and are excited to explore this direction in future work.
>
> Re: few-shot learning vs. OOD detection
>
> We'd like to highlight both the difference and connections between these two learning problems.
>
> **Difference**: In few-shot/zero-shot learning, a model may be given very few or no examples of a specific class in its training data. This class is nonetheless assumed to belong to the training data, and the model is expected to generalize to this class.  In contrast, OOD detection attempts to detect what the model cannot generalize to (e.g., a completely unseen or unknown semantics). From a learning problem perspective, the formulations and objectives are different.
>
> **Connection**: In open-world learning, one can use OOD detection for identifying novel categories as a precursor. Few-shot learning can be useful for subsequent training by incorporating the novel data detected. Though interesting to us, such a learning framework is out of the scope of current work.

---

### Official Review · AnonReviewer4 · 2020-10-30
**an interesting paper studying an underexplored problem**

**Rating:** 7
**Confidence:** 5

**Review:**

======== original feedback ===================

Review: This paper studies out-of-distribution (OOD) detection for multi-label classification with energy-based models. Specifically, the paper proposes to use SumEnergy, which aggregates energy scores from multiple labels, to estimates the OOD. Empirical studies are performed to validate the proposed framework on several benchmarks.

Strength:
+ The paper studies out-of-distribution detection for multi-label classification by using an energy-based framework, which is a real and practical problem.
+ The paper provides comprehensive experiments to show the effectiveness of the proposed energy-based framework for out-of-distribution detection with multi-label classification. The method also establishes the state-of-the-art performance, which is good.

Concerns:
+ since the paper for energy-based out-of-distribution detection has existed. Generalizing it to multi-label context is incremental.
+ missing some important references:  the current paper didn’t discuss and cite [1], which is the first paper to show that a discriminative classifier can be interpreted from an energy-based perspective.
+ the related work about EBM is not comprehensive. Even though the current paper is about discriminative EBM, a discussion about the development of the generative EBMs is desirable.

[1] A Theory of Generative ConvNet (ICML 2016)

======== score changed ==============

My major concern has been addressed by the reply from the authors.  The revised paper has been improved.

---

> ### Author Response · Authors · 2020-11-12
> **[draft updated] Missing references have been added; novelty recap**
>
> We appreciate the reviewer for finding our work interesting and pointing out the missing references! We’ve updated our draft, added citation [1], and several subsequent works in the development of EBMs. **See Page 8-9 for changes**.
>
> Re: Novelty
>
> While the energy score was used for OOD detection in the multi-class setting (Liu et al., 2020), this method unfortunately does not directly generalize to a multi-label classification setting. A multi-label setting in fact exacerbates the difficulty of OOD detection since labels are not mutually exclusive. The key technical challenge and contribution of our work are to derive statistics by considering the joint information across labels. This joint estimation of the OOD score from multiple labels was previously unexplored in literature, hence not incremental but new to our work.
>
> In particular, we showed in Section 2.2 the novel definition of SumEnergy derived from a multi-label classification network, as well as its mathematical interpretations from a joint likelihood view---both of which are new contributions to this work, and are methodologically distinct from prior works.
>
> Our empirical results in Table 1 support precisely the importance of taking into account joint information from across labels, which was not considered in any of the previous baselines (Hendrycks et al. 2019, Liang et al. 2018, Lee et al. 2018) relying on maximum scores.
>
> In summary, our work contributes to the field by both studying an **underexplored problem** (OOD detection for multi-label classification) and proposing an **unexplored solution space** (OOD scoring function estimated jointly from multiple labels). Furthermore, it opens up an **interesting and promising** future direction for OOD detection to consider information jointly from across semantics.

---

> > ### Comment · AnonReviewer4 · 2020-11-17
> > **agree on the novelty of the paper**
> >
> > Thank you for clarifying the difference between the current work and the prior one, and highlighting the novelty. Your reply about the novelty has addressed my concern now. I agree now on that the current paper under review is studying an underexplored problem, which is the OOD detection for multi-label classification. Also, the current revision that takes all the suggestions from the reviewers has greatly improved the paper. Thank you for your efforts.

---

> > > ### Author Response · Authors · 2020-11-17
> > > **thank you!**
> > >
> > > We are really glad to hear the concerns have been addressed.
> > >
> > > Once again thank you for the great comments which helped us improve the work!

---

> > ### Comment · AnonReviewer4 · 2020-11-17
> > **rating is changed**
> >
> > Since the reply has addressed my concern and a revision has made to improve the paper. After I  went over other reviewers' feedbacks, I re-evaluated the current revised paper. I have increased my rating accordingly (from 5 to 7).

---

### Author Response · Authors · 2020-11-17
**Revision summary**

We thank all the reviewers for their insightful feedback. We have additionally responded to each reviewer in detail. We are encouraged that they find our idea to be _interesting_ (R4), _novel_ (R4), _motivated by a very important and underexplored problem_ (R2, R4), and _clearly presented and written_ (R1, R3). We are equally glad they found our results and evaluations _extensive_ (R2), _promising_ (R1), _convincing_ (R3), and _outperforming the state-of-the-art_ (R3, R4). Moreover, we are pleased that (R3) appreciated our mathematical insights on why SumEnergy is suitable for OOD detection in a multi-label setting. **We have incorporated the feedback from all reviewers in our updated draft**. Below we summarize our changes.

[-->R1, R2] We added AUROC curves in Figure 2 (Page6), which shows the effect of threshold $\tau$. We follow the common practice in literature and select $\tau$ based on 95% of true positive rate (fraction of in-distribution images that are correctly classified). Threshold-independent metric AUROC was also provided in Table 1.

[-->R1] We expanded the mathematical interpretation and proof in form of joint conditional likelihood (see Page 4 Equation 12-16, Section 2.2). Notations throughout the paper have been revised as suggested.

[-->R2] We have highlighted the connections to attention-based and graph-based multi-label classification methods  (see Page 4 in Section 2.2).

[-->R3] We expanded the discussion on SumEnergy vs. SumProb, highlight the merits of our method both empirically and theoretically (see Page 7).

[-->R3, R4] We expanded our related work, added missing references on energy-based learning (see Page 8-9).

[-->R3] We have added clear citations to the existing baseline methods in Table 1.

In summary, we believe our work contributes to the field by studying an **underexplored problem** (OOD detection for multi-label classification), proposing an **unexplored and effective solution space** (OOD scoring function estimated jointly from multiple labels, with up to 10.05% FPR reduction compared to the previous best method) that establishes state-of-the-art performance. We also provide theoretical interpretations. Furthermore, our work opens up an **interesting and promising** future direction for OOD detection to consider information jointly from across semantics.

We thank the reviewers again for the constructive comments which helped greatly improve our work!

---

### Decision · Program_Chairs · 2021-01-07
**Final Decision**

**Decision:**

Reject

**Comment:**

The paper aims to do out-of-distribution (OOD) detection in multi-label classification. However,  the challenges of extending energy-based  OOD methods in multiclass to multi-label setting is not big. This paper just defines the label-wise free energy. The key challenging issues in MLC is the label dependency. The paper did not consider modeling the label dependency and devide it to several binary classification issues. And the paper did not provide the theory gurantee. The paper is far below the bar of top conferences.